

# Comparative analysis of rhizosphere soil between three plantation types in Karst Rocky Desertification area by widely targeted metabolomics

Tongtong Guo[*], Ninghua Zhu[*], Ziqian Pan and Peng Dang

College of Forestry, Central South University of Forestry and Technology, Changsha, China
[*] These authors contributed equally to this work.

## ABSTRACT

Understanding the differences in rhizosphere soil microbial metabolites between severely and mildly rocky desertified areas is crucial for developing ecological restoration strategies and land management measures in rocky desertification regions. This study systematically analyzed the differences in rhizosphere soil microbial metabolites of *Toona sinensis*, *Vernicia fordii*, and *Cornus wilsoniana* in severely and mildly rocky desertified areas of Western Hunan using untargeted metabolomics. The results showed that the types and quantities of primary and secondary metabolites in the rhizosphere soil of severely rocky desertified areas were significantly lower than those in mildly rocky desertified areas. Additionally, under severe rocky desertification conditions, 15 common compounds (*e.g.*, 17a-estradiol, adenine, all-trans-retinoic acid) were significantly increased in the rhizosphere soil microbial metabolites of the three tree species. These compounds may provide defense mechanisms for plants to adapt to harsh environments. KEGG metabolic pathway analysis revealed that under severe rocky desertification conditions, *Toona sinensis*, *Vernicia fordii*, and *Cornus wilsoniana* shared six enriched pathways, which play an important role in the biosynthesis of compounds such as phenylpropanoids and unsaturated fatty acids. By revealing the differences in rhizosphere soil microbial metabolites, this study not only deepens the understanding of rocky desertification ecosystems but also provides valuable scientific evidence for ecological restoration and sustainable land management.

## INTRODUCTION

Rocky desertification is a severe land degradation process characterized by soil erosion and bedrock exposure, making it one of the most pressing environmental challenges globally (*Xiong et al., 2009*). In southwestern China, approximately 175,864 km$^2$ of karst areas are affected by this issue (*Juanli et al., 2003*), which also contributes to frequent geological disasters (*Ma et al., 2015*; *Wang, 2003*). This degradation exacerbates local poverty, damages the environment, and significantly hampers socio-economic development (*Chong et al., 2021*; *Chen, 2021*). Vegetation restoration is one of the most commonly used strategies to

Corresponding author
Ninghua Zhu,
zhuninghua@yahoo.com

reduce soil erosion and control rocky desertification. Due to prolonged severe soil erosion, soils in rocky desertification areas are often highly degraded, characterized by thin soil layers, inadequate water supply, and low fertility (*Yan et al., 2020*). Microorganisms play a crucial role in the ecological restoration of rocky desertification areas by influencing tree growth and adaptation. Different vegetation types and management practices have significant impacts on soil microbial communities and ecological functions (*Li et al., 2024*).

Soil microorganisms produce various secondary metabolites, such as antibiotics, antifungal agents, and siderophores, which mediate communication, competition, and interactions with other organisms (*Crits-Christoph et al., 2018*). The rhizosphere, a metabolically active zone, releases bioactive metabolites that influence soil microbial communities more directly than soil factors themselves. These plant and microbial metabolites are critical in regulating biological processes and interactions with neighboring organisms (*Bi et al., 2021*). For example, *Per et al. (2017)* found that the accumulation of proline is a significant metabolic response of plants to salt and drought stress, while betaine can prevent various abiotic stresses by stabilizing protein quaternary structures.

The goal of rocky desertification control and ecosystem restoration is to implement control measures and restoration strategies to maximize the utilization of land and water resources (*Jiang, Lian & Qin, 2014*). Studies indicate that microbial diversity and bacterial metabolic functions vary with different vegetation and land use patterns in rocky desertification areas (*Tang et al., 2024*). Different tree species selection and management practices significantly impact ecological restoration in these areas. Choosing appropriate tree species and optimized management practices can effectively improve soil quality, promote ecosystem stability, and aid in restoration. There remains a significant research gap in understanding the differences in microorganisms and metabolites among various ecological tree species and their mediated tree-soil feedback and adaptation mechanisms in rocky desertification areas. Although many studies have explored the DNA sequences of microorganisms in desertified soils, research on their metabolic activities is relatively scarce.

In this study, *Toona sinensis*, *Vernicia fordii*, and *Cornus wilsoniana* were selected as representative tree species to explore the differences in rhizosphere soil microbial metabolites under severe rocky desertification (Groups A, C, and E) and mild rocky desertification (Groups B, D, and F) conditions. These tree species were chosen for their prevalence and ecological significance in the local ecosystem. Using untargeted metabolomics analysis, we aim to identify the patterns of soil microbial metabolite changes and assess their potential impact on soil quality and ecological functions. Additionally, the study seeks to reveal the metabolic pathways of rhizosphere microbial communities in different tree species under rocky desertification conditions and how these pathways reflect soil microbial responses to environmental changes. Understanding these microbial metabolic pathways will not only aid in predicting the impact of rocky desertification on soil ecological functions but also provide guidance for developing management practices to restore desertified land and improve soil quality. The findings of this study will offer in-depth insights into the impact of rocky desertification on soil microbial metabolites,

providing a scientific basis for future soil biology research and sustainable land management practices.

## SURVEY METHODOLOGY

### Study site

The experimental site is located in the Qingshan State-owned Forest Farm in the Xiangxi Tujia and Miao Autonomous Prefecture (Wuling Mountain Rocky Desertification Comprehensive Management Long-term National Research Base), specifically in Qingshan Town, Yongshun County, Xiangxi Autonomous Prefecture, Hunan Province (110°13′40.296″E, 29°3′21.59″N). This site is situated in the central region of the Wuling Mountains. The highest elevation reaches 820 m, while the lowest is 320 m. The bedrock is limestone, and the soil is predominantly yellow-brown, characteristic of severely rocky desertified areas. The climate of the area is a mid-subtropical monsoon humid climate, with distinct continental monsoon features. The region receives an average annual precipitation of 1,300–1,500 mm and an average annual temperature of 16.3 °C (*Huang et al., 2021*).

The Autonomous Prefecture Forest Ecology Research Experimental Station covers a total area of 706.4 hectares, with forested land accounting for approximately 679.1 hectares. The standing timber volume is about 70,000 cubic meters. The region is rich in biodiversity, housing over 1,300 species of flora and fauna. The forest coverage rate is 94%, and the greening rate of forested areas is 95.6%.

### Sample collection

A typical plot sampling survey method was adopted, referring to the rocky desertification classification standards (*National Forestry and Grassland Administration of the People's Republic of China, 2009*). Based on factors such as rock exposure rate, vegetation type, comprehensive vegetation cover, and soil layer thickness, rocky desertification was classified into two levels: severe rocky desertification (HRD) and mild rocky desertification (LRD). Three tree species were selected for each level: *Toona sinensis*, *Vernicia fordii*, and *Cornus wilsoniana*. Four replicates were set up for each tree species, resulting in a total of 24 fixed plots, each with a size of 20 m × 20 m. This sample size and the number of replicates were chosen to ensure that sufficient statistical power was achieved to detect significant differences between the two levels of rocky desertification. The number of replicates per species and the total number of plots (24) were based on statistical justification. The plots were marked as Group A and Group B (*Toona sinensis* HRD and LRD), Group C and Group D (*Vernicia fordii* HRD and LRD), and Group E and Group F (*Cornus wilsoniana* HRD and LRD).

Prior to formal sampling, a field survey was conducted to determine the direction and location of rhizosphere growth. Using sterile tools, rhizosphere soil (root zone soil) and its surrounding soil were excavated. For each plot, the rhizosphere soil from three randomly selected trees was mixed to increase the representativeness of the samples, resulting in a total of 72 mixed soil samples. This approach of combining soil samples from multiple trees ensures that the samples accurately reflect the variability within the study area. All

samples were placed in clearly labeled ziplock bags, preserved on dry ice, and immediately transported back to the laboratory for subsequent processing.

In the laboratory, visible plant debris, stones, and other impurities were removed from the samples. The soil was then sieved through a two mm mesh to eliminate larger particles. The resulting fine soil was used for subsequent microbial and chemical analyses. Portions of the soil samples designated for microbial analysis were placed in 15ml centrifuge tubes and immediately frozen for microbial metabolite analysis.

## Materials and methods
### Sample preparation
First of all, an appropriate amount of sample needs to be weighed extremely accurately and carefully placed into a two mL centrifuge tube. Then, 600 µL of methanol solution of 2-chloro-L-phenylalanine with a concentration of four ppm was accurately added. After that, vortex oscillation immediately performed for 30 s to ensure that the sample and the solution are fully mixed.

Subsequently, a steel ball was added into the centrifuge tube and then the entire centrifuge tube was placed in a tissue grinder. It was ground at a frequency of 55 Hz for 90 s to fully grind and refine the sample.

After that, ultrasonic treatment was conducted on the mixture in the centrifuge tube at room temperature for 30 min. After the ultrasonic treatment was completed, it was placed on ice and let stand for 30 min to further stabilize the mixture in a low-temperature environment.

Finally, it was centrifuged at a speed of 12,000 rpm under a low temperature of 4 °C for 10 min. After centrifugation, the supernatant was removed and filtered through a 0.22 µm filter membrane. The filtered liquid was carefully added into the detection bottle for use in liquid chromatography-mass spectrometry (LC-MS) detection (*Vasilev et al., 2016*).

### Liquid chromatography conditions
The Thermo Vanquish ultra-high performance liquid system uses an ACQUITY UPLC® HSS T3 (2.1 × 100 mm, 1.8 µm) chromatographic column. The system is set with a flow rate of 0.3 mL/min, a column temperature of 40 ° C, and an injection volume of two µL. In positive ion mode, the mobile phase is composed of 0.1% formic acid acetonitrile (denoted as B2) and 0.1% formic acid water (denoted as A2). The gradient elution program is as follows: from 0 to 1 min, the proportion of B2 is 8%; from 1 to 8 min, the proportion of B2 rises from 8% to 98%; from 8 to 10 min, the proportion of B2 is 98%; from 10 to 10.1 min, the proportion of B2 drops from 98% to 8%; from 10.1 to 12 min, the proportion of B2 is 8%. In negative ion mode, the mobile phase is acetonitrile (denoted as B3) and 5 mM ammonium formate water (denoted as A3). The gradient elution program is: from 0 to 1 min, the proportion of B3 is 8%; from 1 to 8 min, the proportion of B3 rises from 8% to 98%; from 8 to 10 min, the proportion of B3 is 98%; from 10 to 10.1 min, the proportion of B3 drops from 98% to 8%; from 10.1 to 12 min, the proportion of B3 is 8% (*Zelena et al., 2009*).

### Mass spectrum conditions

The Thermo Orbitrap Exploris 120 mass spectrometer detector, equipped with an electrospray ionization source (ESI), collects data in positive and negative ion modes respectively. In the positive ion mode, the spray voltage is 3.50 kV; in the negative ion mode, the spray voltage is −2.50 kV. The sheath gas is 40 arb and the auxiliary gas is 10 arb. The capillary temperature is set at 325 °C. A full scan was performed at the first level with a resolution of 60,000. The first-level ion scanning range is m/z 100 to 1,000. At the same time, high-energy collision dissociation (HCD) was used for secondary fragmentation. The collision energy is 30%, and the secondary resolution is 15,000. The first four ions were fragmented before the signal was collected, and dynamic exclusion was used to remove unnecessary MS/MS information (*Want et al., 2013*).

## Data processing and analysis

The raw data were firstly converted to mzXML format by MSConvert in ProteoWizard software package (v3.0.8789) (*Rasmussen et al., 2022*) and processed using XCMS (*Navarro-Reig et al., 2015*) for feature detection, retention time correction and alignment. The metabolites were identified by accuracy mass (<30 ppm) and MS/MS data which were matched with HMDB (*Wishart et al., 2007*), massbank (*Horai et al., 2010*), LipidMaps (*Manish et al., 2007*), mzcloud (*Abdelrazig et al., 2020*), KEGG (*Ogata et al., 1999*), The robust LOESS signal correction (QC-RLSC) (*Gagnebin et al., 2017*) was applied for data normalization to correct for any systematic bias. After normalization, only ion peaks with relative standard deviations (RSDs) less than 30% in QC were kept to ensure proper metabolite identification.

Ropls (*Thévenot et al., 2015*) software was used for all multivariate data analyses and modelings. Data were mean-centered using scaling. Models were built on principal component analysis (PCA) and partial least-square discriminant analysis (OPLS-DA).

Differential metabolites were subjected to pathway analysis by MetaboAnalyst (*Xia & Wishart, 2011*), which combines results from powerful pathway enrichment analysis with the pathway topology analysis. The identified metabolites in metabolomics were then mapped to the KEGG pathway for biological interpretation of higher-level systemic functions. The metabolites and corresponding pathways were visualized using KEGG Mapper tool.

# RESULTS

## Principal component analysis

Principal component analysis (PCA) was used to observe the trend of separation between treatments and the presence of outliers in the experiment, and to reflect assess the variation between and within treatments based on the original data. In this study, positive ion mode and negative ion mode were used respectively, and the treatment methods were divided into two principal components (PCs). In the positive ion mode (Fig. 1A), PC1 and PC2 explained 14.9% and 11.7% of the total variance, respectively. Distinct clustering of treatment groups was observed, with groups A, B, and C separated along the PC1 axis, suggesting significant metabolic differences between these treatments. In contrast, groups D
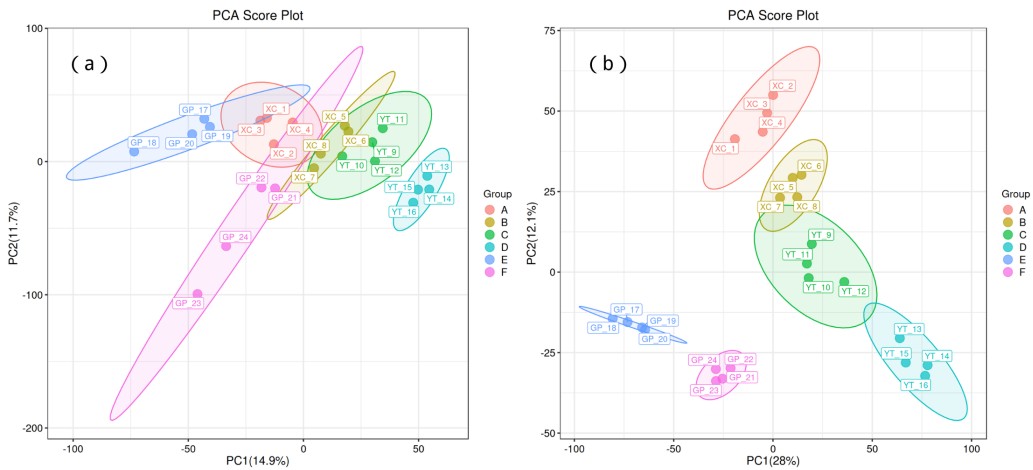

**Figure 1** **Principal component analysis (PCA) of rhizosphere soil microbial metabolites of *Toona sinensis*, *Vernicia fordii*, and *Cornus wilsoniana* under different degrees of rocky desertification.** (A) Positive ion mode; (B) negative ion mode.

and E displayed partial overlap, indicating some shared metabolic characteristics. Sample replicates within each group were tightly clustered, demonstrating high experimental consistency and repeatability. Similarly, in the negative ion mode (Fig. 1B), PC1 and PC2 explained 28% and 12.1% of the total variance, respectively. Groups A and B showed clear separation along PC1, while groups C and D exhibited distinct clustering along PC2, reflecting treatment-induced metabolic divergence.

## Orthogonal partial least squares discriminant analysis

To obtain a high level of treatment separation and obtain a better understanding of variables responsible for classifcation, orthogonal partial least squares discriminant analysis (OPLS-DA) was applied. In this analysis, T scores in positive ion mode (10.3%) indicate predicted PC scores for PC1, and orthogonal T scores (12.3%) indicate orthogonal PC scores (Fig. 2A). This revealed clear group separation, with groups A, B, and C distinctly clustered. In negative ion mode (Fig. 2B), T-scores (13.7%) indicate predicted PC scores for PC1, while orthogonal T-scores (26.3%) show orthogonal PC scores, demonstrating pronounced separation between groups E and F along PC2. Based on the model validation, R2 (model explanatory rate) and Q2 (model predictive power) were 0.998 and 0.732, respectively, in the positive ion mode, and 0.989 and 0.912, respectively, in the negative ion mode, and $p < 0.005$, indicating that the measured data were reliable.

Key metabolites in positive ion mode, such as M387T40 and M501T573, significantly contribute to the differences between groups (Fig. 2C); important metabolites in negative ion mode, such as M419T470_3 and M427T594, also show significant contributions (Fig. 2D).

## Differential metabolite analysis

In this study, a total of 11,085 primary metabolites were detected in the positive ion mode. Compared to the metabolites under LRD conditions, Group A had 454 upregulated and

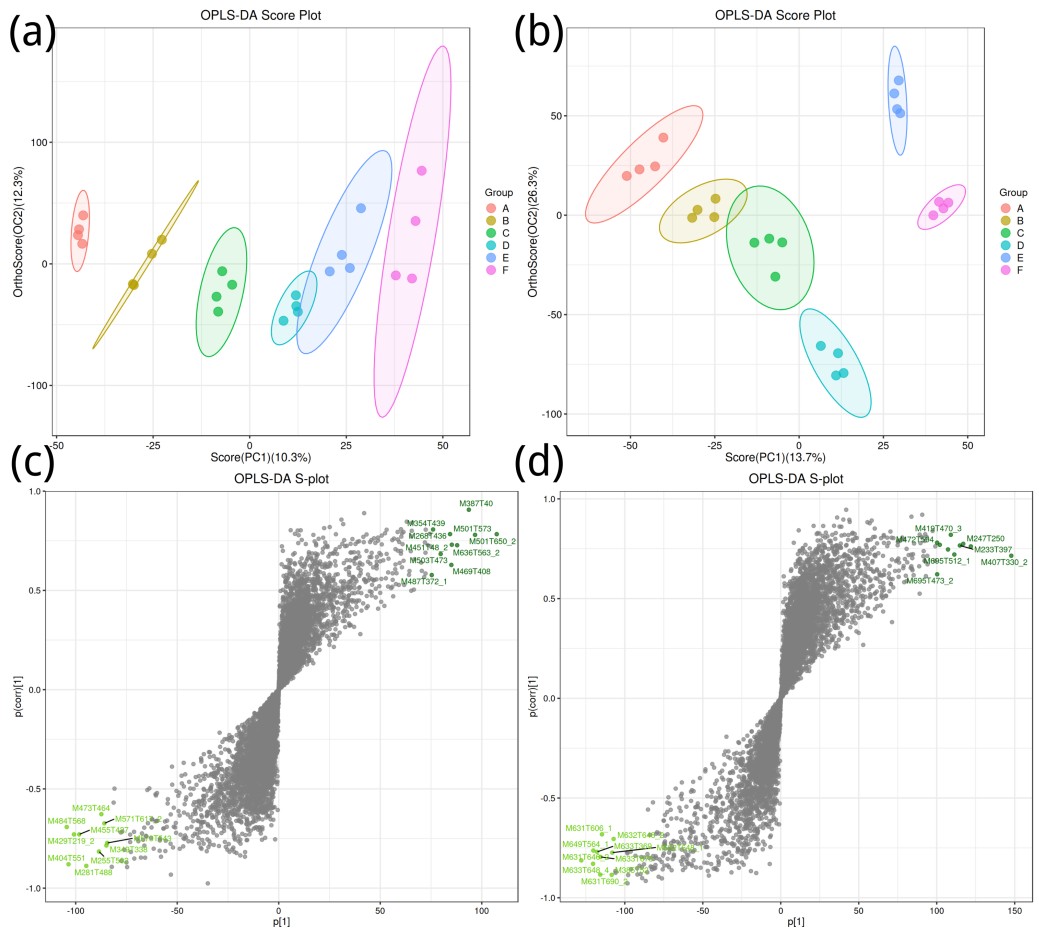

**Figure 2** **Orthogonal partial least squares discriminant analysis plot.** (A, B) Positive ion mode; (B, D) negative ion mode.

1,003 downregulated metabolites, Group C had 404 upregulated and 1,165 downregulated metabolites, and Group E had 348 upregulated and 1,621 downregulated metabolites (Table 1). In the negative ion mode, a total of 10,020 primary metabolites were detected. Compared to the number of metabolites under LRD conditions, Group A had 524 upregulated and 875 downregulated metabolites, Group C had 356 upregulated and 1,899 downregulated metabolites, and Group E had 300 upregulated and 2,064 downregulated metabolites (Table 1). Additionally, a total of 4,710 secondary metabolites were detected. Compared to the number of metabolites under LRD conditions, Group A had 20 upregulated and 31 downregulated metabolites, Group C had 13 upregulated and 51 downregulated metabolites, and Group E had 8 upregulated and 54 downregulated metabolites (Table 1). The results indicate that under HRD conditions, the number of primary metabolites is significantly lower than under LRD conditions. Secondary metabolites also show a predominantly downregulated trend under KRD conditions, indicating that the KRD environment has a significant inhibitory effect on the metabolic activity of soil microbial communities.
**Table 1** Statistics of differential metabolites in positive ion mode, negative ion mode, and MS/MS secondary analysis.

| Comparison | Positive ion mode | | | | Negative ion mode | | | | (MS/MS secondary analysis) | | | |
|---|---|---|---|---|---|---|---|---|---|---|---|---|
| | Total | Up | Down | Total_DE | Total | Up | Down | Total_DE | Total | Up | Down | Total_DE |
| B vs A | 8,899 | 454 | 1,003 | 1,457 | 7,264 | 524 | 875 | 1,399 | 315 | 20 | 31 | 51 |
| D vs C | 8,899 | 404 | 1,165 | 1,569 | 7,264 | 356 | 1,899 | 2,255 | 315 | 13 | 51 | 64 |
| F vs E | 8,899 | 348 | 1,621 | 1,969 | 7,264 | 300 | 2,064 | 2,364 | 315 | 8 | 54 | 62 |

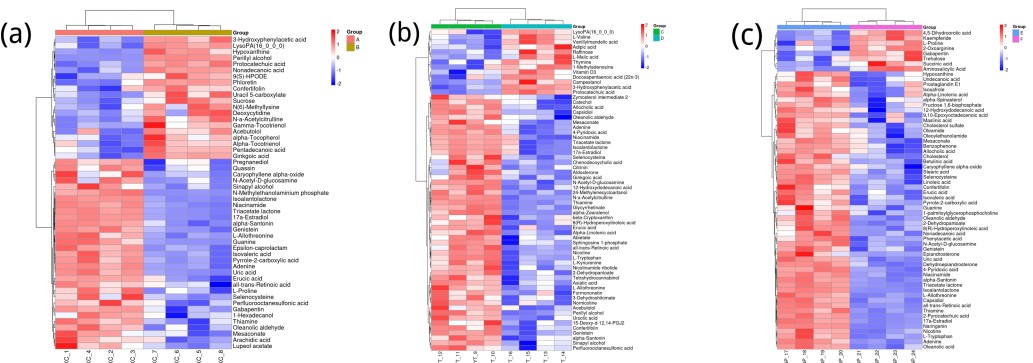

**Figure 3** Hierarchical cluster analysis of metabolites in the rhizosphere soil of *Toona sinensis*, *Vernicia fordii*, and *Cornus wilsoniana* under HRD and LRD conditions. (A) *Toona sinensis*. (B) *Vernicia fordii*. (C) *Cornus wilsoniana*.

## Metabolite heatmap analysis

Figure 3A shows the differences in metabolite abundance in the rhizosphere soil of *Toona sinensis* under HRD (Group A) and LRD (Group B) conditions. The results indicate that under LRD conditions, the abundance of compounds such as 3-hydroxyphenylacetic acid, LysoPA (16:0/0:0), hypoxanthine, and other organic acids, phospholipids, and bases significantly increased. Conversely, under HRD conditions, the abundance of Isovaleric acid, N-methylethanolaminium phosphate, isoalantolactone, and other organic acids, amino acids, and lactones was higher.

Figure 3B shows the differences in metabolite abundance in the rhizosphere soil of *Vernicia fordii* under HRD conditions (Group C) and LRD conditions (Group D). The study found that under LRD conditions, the abundance of protocatechuic acid, 3-hydroxyphenylacetic acid, campestanol, and other organic acids and sterols significantly increased, indicating their important role in adapting to environmental stress. In contrast, under HRD conditions, the abundance of 12-hydroxydodecanoic acid, N-a-acetylcitrulline, 17a-estradiol, and other organic acids, amino acids and derivatives, and steroids was higher.

Figure 3C shows the differences in metabolite abundance in the rhizosphere soil of *Cornus wilsoniana* under HRD conditions (Group E) and LRD conditions (Group F). The results indicate that under LRD conditions, the abundance of 4,5-dihydroorotic acid, aminosalicylic acid, Kaempferide and other organic acids and phenolic compounds significantly increased. Conversely, under HRD conditions, the abundance of Oleanolic
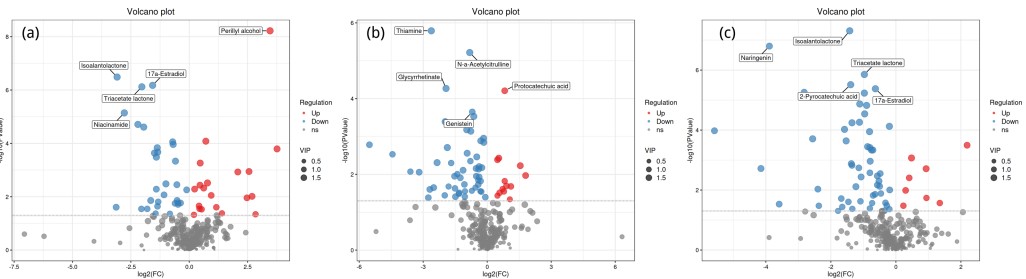

**Figure 4** Volcano plots of significant differential expression of metabolites in the rhizosphere soil of *Toona sinensis*, *Vernicia fordii*, and *Cornus wilsoniana* under HRD and LRD conditions. (A) *Toona sinensis*. (B) *Vernicia fordii*. (C) *Cornus wilsoniana*. Red dots represent upregulated metabolites, blue dots represent downregulated metabolites, and grey dots represent non-significant metabolites. Metabolites with higher VIP values are indicated with larger dots.

acid, cholesterol sulfate, L-allothreonine, and other organic acids, steroids, and amino acids and derivatives was higher.

Figure 3 also demonstrates a clear global pattern in the grouping of metabolite abundance, as shown by the color scale in the heatmaps. The color groupings suggest a global trend in metabolite composition, highlighting distinct differences in metabolite abundance across the different environmental conditions (HRD *vs.* LRD). The observed patterns indicate that under severe rocky desertification (HRD), 17a-estradiol, adenine, all-trans-retinoic acid, alpha-santonin, erucic acid, genistein, isoalantolactone, L-allothreonine, mesaconate, N-Acetyl-D-glucosamine, niacinamide, oleanolic aldehyde, selenocysteine, thiamine, and triacetate lactone tend to increase significantly across all species, reflecting a possible common metabolic response to harsh environmental stress. In contrast, under mild rocky desertification (LRD), the patterns show a distinct shift in metabolite composition, suggesting an adaptive strategy for surviving under less stressful conditions. These findings reinforce the idea that the metabolic responses of *Toona sinensis*, *Vernicia fordii*, and *Cornus wilsoniana* are influenced by the severity of the environmental stress, which could be generalized to similar karst ecosystems.

## Volcano plot analysis

Figure 4 shows the significant differential expression of metabolites in the rhizosphere soil of *Toona sinensis*, *Vernicia fordii*, and *Cornus wilsoniana* under HRD and LRD conditions. The results indicate that under HRD conditions, the abundance of certain metabolites, such as perillyl alcohol and protocatechuic acid, significantly increased, suggesting that they may play a crucial role in responding to environmental stress and maintaining soil microbial community functions. Conversely, under LRD conditions, other metabolites, such as 17a-estradiol and isoalantolactone, showed higher abundances, indicating their critical importance in maintaining the balance and function of microbial communities under LRD conditions.

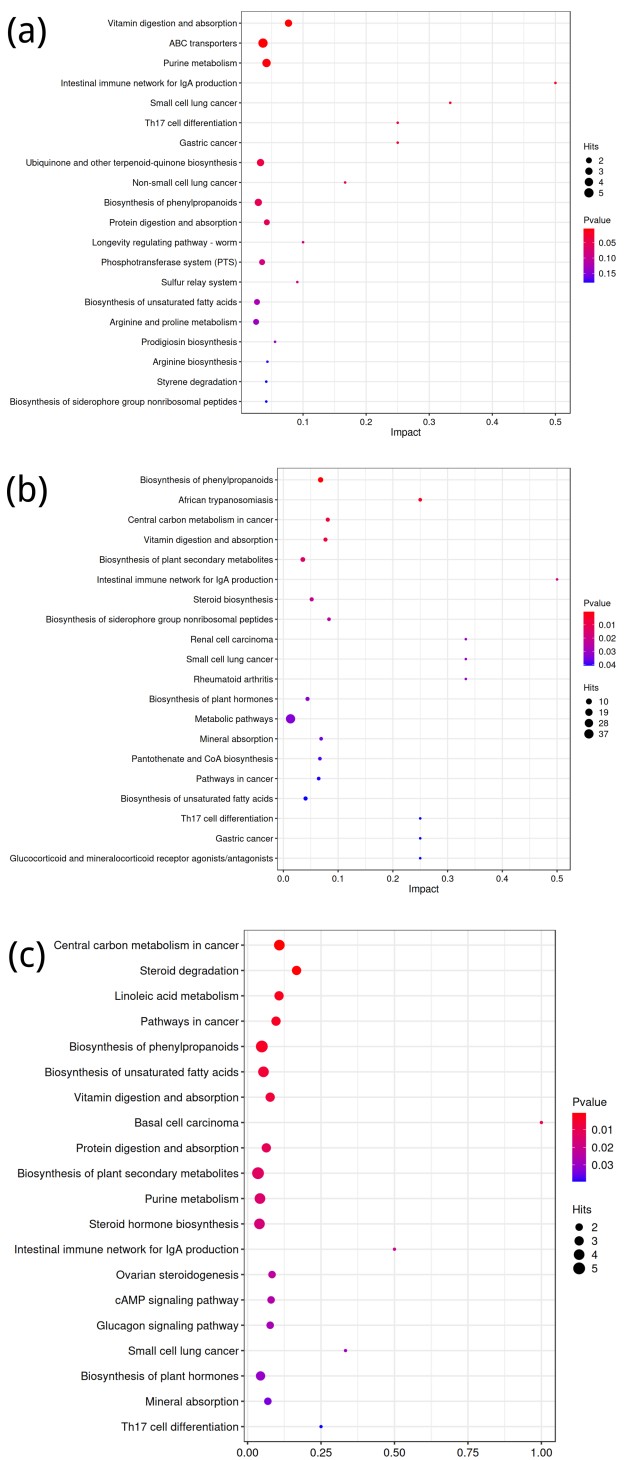

**Figure 5** KEGG metabolic pathway analysis of significant differential expression of metabolites in the rhizosphere soil of *Toona sinensis*, *Vernicia fordii*, and *Cornus wilsoniana* under HRD and LRD conditions. (A) *Toona sinensis*. (B) *Vernicia fordii*. (C) *Cornus wilsoniana*.

**Metabolic pathway analysis (KEGG) of differential metabolites**

Using the KEGG database, differential metabolites were annotated to identify metabolic pathway differences between groups A and B, C and D, and E and F. For the comparison between groups A and B, the top five differential metabolic pathways were related to Vitamin digestion and absorption, ABC transporters, purine metabolism, the Intestinal immune network for IgA production, and small cell lung cancer (Fig. 5A). While the latter pathway might seem unexpected in the context of our study, it likely reflects broader immune and metabolic alterations associated with the experimental conditions.

For the comparison between groups C and D, key pathways included the biosynthesis of phenylpropanoids, African trypanosomiasis, central carbon metabolism in cancer, vitamin digestion and absorption, and the biosynthesis of plant secondary metabolites (Fig. 5B). These pathways align with metabolic processes that may be influenced by both external stressors and genetic factors in the studied species.

In the comparison between groups E and F, the top five differential metabolic pathways involved central carbon metabolism in cancer, steroid degradation, linoleic acid metabolism, pathways in cancer, and biosynthesis of phenylpropanoids (Fig. 5C). These pathways are consistent with known alterations in metabolic activity and suggest potential implications for the broader physiological responses observed in this group.

## DISCUSSION

The karst ecosystem is one of the most fragile terrestrial ecosystems, characterized by a slow soil formation process and low forest coverage rates (*Zhao et al., 2017*; *Tang et al., 2018*; *Gao et al., 2023*; *Li et al., 2021*). Vegetation restoration in rocky desertification areas is a key strategy for advancing ecological restoration and management in these regions (*Liu et al., 2023*). Soil microorganisms play a crucial role in this process, directly or indirectly influencing soil nutrient cycling, soil structure, and biogeochemical cycles (*Zheng et al., 2020*; *Jin et al., 2019*; *Yuguang et al., 2014*).

Microbial metabolites play an essential role in microbial metabolism, facilitating biotransformation processes and the degradation of xenobiotic compounds. Some metabolites, such as organic acids, also act as plant growth-promoting factors, promoting plant resilience in stressful environments (*Minhas et al., 2024*). Plants alter soil microbial communities by secreting active molecules into the rhizosphere, including primary and secondary metabolites (*Schütz et al., 2021*; *Lynch & Whipps, 1990*; *Park, Hochholdinger & Gierl, 2004*; *Hartmann et al., 2009*; *Cesco et al., 2010*). These metabolites affect soil properties and, consequently, the growth of subsequent plant generations, highlighting the importance of metabolites in regulating plant-associated microbial communities and establishing the key role of plant secondary metabolites in rhizosphere processes (*Peiffer et al., 2013*; *Lundberg et al., 2012*; *Horton et al., 2014*; *Edwards et al., 2015*; *Lebeis et al., 2015*; *Wagner et al., 2016*). Different plants exhibit significant differences in the secretion of bioactive metabolites, leading to varying soil feedback effects (*Klironomos, 2002*). Therefore, enhancing the abundance of advantageous metabolites in the rhizosphere of tree species in rocky desertification areas could improve tree survival and growth under harsh conditions, thereby achieving the goal of vegetation restoration in these areas.

This study systematically analyzed the differences in microbial metabolites in the rhizosphere soil of *Toona sinensis*, *Vernicia fordii*, and *Cornus wilsoniana* in severely and mildly rocky desertified areas of western Hunan using non-targeted metabolomics. The results showed that severe rocky desertification significantly affected the primary and secondary metabolites in the rhizosphere soil. Regarding primary metabolites, the variety and quantity in severely rocky desertified areas were significantly lower than in mildly rocky desertified areas, indicating that severe rocky desertification significantly inhibits the metabolic activity of soil microbial communities. This is consistent with the findings of *Xie et al. (2015)* who reported a sharp decline in soil microbial activity with increasing desertification severity (*Lianwu et al., 2015*; *Banning et al., 2011*). Heatmap results showed an increase in organic acids such as all-trans-retinoic acid, Mesaconate, Erucic acid, Oleanolic aldehyde, L-allothreonine, and selenocysteine under severe rocky desertification conditions, consistent with the findings of *Tang et al. (2024)* and *Song et al. (2012)* observed an increase in organic acids in the rhizosphere of locust trees under strong KRD conditions, and *Song et al. (2012)* found that drought stress increased organic acids such as malic acid in the plant rhizosphere. Besides these six organic acids, other metabolites like 17a-estradiol, adenine, alpha-santonin, genistein, isoalantolactone, N-Acetyl-D-glucosamine, niacinamide, thiamine, and triacetate lactone also significantly increased. Therefore, we speculate that the increase in these 15 rhizosphere soil microbial metabolites might provide *Toona sinensis*, *Vernicia fordii*, and *Cornus wilsoniana* with defense mechanisms, helping them adapt to harsh environmental conditions.

KEGG is a powerful tool for metabolic analysis and research, providing pathways for plant secondary metabolism and genomes of plants and related organisms (*Zhou et al., 2018*; *Zhou et al., 2016*; *Kanehisa, 2016*). According to KEGG results, under severe rocky desertification conditions, *Toona sinensis*, *Vernicia fordii*, and *Cornus wilsoniana* shared the following six enriched pathways: biosynthesis of phenylpropanoids, biosynthesis of unsaturated fatty acids, Intestinal immune network for IgA production, Small cell lung cancer, Th17 cell differentiation, and vitamin digestion and absorption. These common enriched pathways indicate that these tree species adopt similar metabolic strategies under severe rocky desertification conditions, reflecting their physiological adaptation mechanisms to harsh environments. Previous research found that a glycine-rich RNA-binding protein, OsGRP3, in rice enhances its drought resistance by altering the phenylpropanoid biosynthesis pathway (*He et al., 2020*). It is thus speculated that the growth of *Toona sinensis*, *Vernicia fordii*, and *Cornus wilsoniana* in severely rocky desertified areas may also rely on the biosynthesis pathway of phenylpropanoids. Additionally, studies have shown that unsaturated fatty acids play multiple important roles in plants, such as participation in biosynthesis and regulation (*Xu et al., 2022*). These mechanisms might be crucial in helping these tree species adapt to rocky desertification environments. Furthermore, the related metabolic pathways provide a theoretical basis for further exploration of the biosynthesis of metabolites related to *Toona sinensis*, *Vernicia fordii*, and *Cornus wilsoniana*.

Through this study, we found a significant impact of HRD on the microbial metabolites in the rhizosphere soil. However, relying solely on the changes in the abundance of

microbial metabolites is insufficient to comprehensively explain the complex mechanisms in vegetation restoration. Due to factors such as region and environment, the applicability of the research findings to other rocky desertification areas may be limited.The next step is to integrate the changes in plant root characteristics, microbial community structure, and soil physicochemical properties to jointly assess their combined effects on plant growth and ecological restoration. Such a comprehensive analysis will provide more actionable strategies to enhance the effectiveness of ecological restoration.

## CONCLUSIONS

This study analyzed the differences in microbial metabolites in the rhizosphere soil of *Toona sinensis*, *Vernicia fordii*, and *Cornus wilsoniana* in severely and mildly rocky desertified areas of western Hunan. The results indicated that rocky desertification significantly inhibited the metabolic activities of soil microorganisms, reducing the variety and quantity of metabolites. At the same time, six organic acids, including all-trans-retinoic acid, mesaconate, erucic acid, oleanolic aldehyde, L-allothreonine, and selenocysteine, as well as nine other rhizosphere soil microbial metabolites, such as 17a-estradiol, significantly increased under severe rocky desertification conditions, helping plants adapt to harsh environments. Metabolic pathway analysis showed that the three tree species adopted six common enriched pathways under severe rocky desertification conditions. Under HRD conditions, multiple organic acids and other metabolites significantly increased. These metabolites may provide plants with defense mechanisms to adapt to harsh environments. Future research could further explore how these metabolic mechanisms promote plant recovery and how these findings can be applied in ecological restoration projects.

### Funding
This work was funded by The Forestry Department of Hunan Province (XLKY202330). The funders had no role in study design, data collection and analysis, decision to publish, or preparation of the manuscript.

### Grant Disclosures
The following grant information was disclosed by the authors:
The Forestry Department of Hunan Province: XLKY202330.

### Competing Interests
The authors declare there are no competing interests.

### Author Contributions
- Tongtong Guo conceived and designed the experiments, performed the experiments, analyzed the data, prepared figures and/or tables, authored or reviewed drafts of the article, and approved the final draft.
- Ninghua Zhu conceived and designed the experiments, authored or reviewed drafts of the article, and approved the final draft.
- Ziqian Pan performed the experiments, analyzed the data, prepared figures and/or tables, authored or reviewed drafts of the article, and approved the final draft.
- Peng Dang conceived and designed the experiments, authored or reviewed drafts of the article, and approved the final draft.

## Data Availability

Data is available at metaboLights: MTBLS11482.

## Supplemental Information

Supplemental information for this article can be found online at http://dx.doi.org/10.7717/peerj.19131#supplemental-information.

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
