# Peer review of "Comparative analysis of rhizosphere soil between three plantation types in Karst Rocky Desertification area by widely targeted metabolomics"

_PeerJ, doi:10.7717/peerj.19131_

## Round 0.1 · original submission · Major Revisions

Dear authors,

I ask you to improve the manuscript very carefully in accordance with the recommendations of the reviewers. The direction of research is quite new, so I hope that this article will arouse considerable interest among readers. The authors have a greater responsibility: to present the data in a form that will become standard for many years to come for this direction of research. Control samples should also be analyzed and presented in figures.

·

Basic reporting

The manuscript provides clear and well-organized content, presenting an important study on rhizosphere soil metabolomics in karst rocky desertification areas. The language is generally professional and accessible, though some grammatical revisions could enhance readability, particularly in complex sections like the results. Overall, the paper is structured well, with a logical flow from the introduction to conclusions.

- Introduction and Background: The introduction adequately contextualizes the study, discussing the significance of karst rocky desertification and the roles of soil metabolites and microbial communities.
- Figures and Tables: Figures and tables are relevant, clearly labeled, and add value to the discussion, effectively visualizing key findings in PCA, heatmap analyses, and pathway analysis.

Suggestions:
- Language Refinement: Improving sentence structure in specific sections (e.g., "Discussion" and "Results") would clarify points. Proofreading by a native English speaker or using professional editing services could help.
- Literature Review: A more detailed exploration of recent studies in rhizosphere soil metabolomics and karst ecosystems may strengthen the introduction. This would clarify the study’s unique contribution.

Experimental design

The study aligns with the journal's scope, presenting original research relevant to ecological restoration in fragile karst landscapes. The methods section is generally thorough, allowing for replication and providing a robust framework for the experiment.

- Experimental Design: The study's design of comparing different plantation types across two levels of desertification is well-conceived, addressing an ecological gap and providing insight into plant-soil-microbe interactions.
- Detailed Methodology: The authors provide sufficient technical detail on sample collection, metabolite analysis, and statistical methods (PCA, OPLS-DA, pathway analysis), allowing others to reproduce the findings.

Suggestions:
- Sample Preparation and Replication: While the methods are robust, including a justification for the sample sizes and replication would strengthen the reliability of the findings.
- Although ethical approval is less relevant here, mentioning any environmental protection or sustainability measures taken during sample collection could add transparency, as this area is sensitive to ecological disturbance.

Validity of the findings

The authors present a compelling case for the ecological impact of metabolite variations, offering a comprehensive analysis of primary and secondary metabolites under different desertification conditions. Statistical analyses are robust, with PCA and OPLS-DA used effectively to identify key metabolites and pathways.

- Data Integrity: The study includes ample data points and controls, ensuring a high standard of validity. The pathway analysis adds depth, linking results to broader ecological functions.
- Interpretation of Results: The discussion connects findings to known physiological responses of plants in desertified areas, enhancing understanding of microbial functions in soil health.

Suggestions:
- Data Accessibility: The manuscript could clarify data availability for other researchers. Providing raw data (if not already) would align with open science practices.
- Limitations and Generalizability: While conclusions are supported by results, a section discussing limitations and the specificity of findings to similar karst environments would provide balance.

Additional comments

This manuscript is a significant contribution to karst ecology and restoration science. Its strengths lie in the interdisciplinary approach, integrating metabolomics with ecological restoration needs. However, further clarity on methodological decisions and nuanced discussions of limitations could enhance the manuscript's robustness.

·

Basic reporting

The English writting is good enough. Nonetheless, I would recommend editing it a bit as there is a significant level of redundancy, phrases that are repeated a lot, mostly in the introduction section. I think that making it more concise would improve understanding.

The arguments in the introduction and the problem background help contextualize the presented research approach in a very logical and straightforward way.

However, throughout the work there is no clear comparison between the two conditions (LRD and HRD). It is mentioned in the discussion that there are metabolic differences, but these were never clearly pointed out in the results, beyond lists of compounds. There is no comparative analysis.

The figures are of very poor quality, they cannot be read well and therefore interpreted. In particular, Figure 3 looks very small, it would be difficult for the reader to draw quick conclusions.

In the results section, the findings are very underestimated, there is not much description and no interpretation. The discussion is also very poor, it focuses only on the last results and to only makes some mentions, without much depth or an effort to give a contextualized explanation to the results.

Experimental design

The research question is clearly stated and addresses a little-explored topic, by evaluating the contributions of the microbial communities associated with the roots of three plants at the metabolic level. Which is a clear gap in knowledge, since there are very few works exploring the rhizospheric communities of desertified soils, these generally focus on DNA sequences.

Regarding sampling, there is no mention of whether a protected area was sampled or whether the selected plant species require some type of authorizations, so it is assumed that there are no bioethical or biosecurity issues. The rest of the work does not raise any concerns in this regard.
The sample size meets the minimum required to obtain exploratory conclusions, but not to reach high statistical power.

The methodology is quite clear and straightforward, very detailed processing and analysis methods regarding metabolomics, so I do not see any problems with this being replicated.

Validity of the findings

The results of this research, although exploratory, are quite valuable as they point to key metabolisms that may be important in plants adaptation to desertification and climate change. Thus, guide future studies with a larger number of samples and scope. The conclusions are well supported by experimental results and data, without going beyond the limits with over speculations. All experimental data from the analyses and metabolites measurements are available in the supplementary tables. However, a better interpretation and discussion of the findings is very necessary to achieve the true value of this work.

Additional comments

Scientifict names shoukd be italized

Figure 1 has an error, the axis values do not correspond to what was stated in lines 187-189. I suspect that Figure 1B is incorrect and the authors mistakenly replicated Figure 1A. Because of this it is impossible to correctly interpret these results.

In the results section, figures 1 and 2 are neither described nor interpreted, the authors simply refer to the axes values in the respective analyses, but no mention is made of groups and samples replicates segregation or grouping.

Figure 3 also shows a lot of information that is very neglected in the results section, the authors overly simplify this. There is also no reference to the global patterns that are clearly observed in the grouping of the colors corresponding to the heat scale.
Are there any similarities/differences between the three plants species or between both conditions (LRD and HRD)?

Please fix line 255, is something odd with it.

In general, KEGG pathway analyzes must be meticulously reviewed and in some cases manually curated. The identification of pathways such as Intestinal immune network for IgA production, Small cell lung cancer, and Th17 cell differentiation. This did not call the authors attention? Because this doesn't sound very logical and if there was any explanation for this, it was totally ignored.

Reviewer 3 ·

Basic reporting

1. I recommend writing in the third person
2. L. 122 references should cite properly. Please check the journal guideline for the same
3. The manuscript seems to me that many paragraphs need to be polished.
4. Nothing professional, subheading should be differentiated

Experimental design

1. The study is designed to decode metabolomics of rhizosphere soils; however, I could not find control soil samples in any of the datasets

Validity of the findings

1. L. 150 Rewrite the sentences
2. State data availability statement

Additional comments

no comment

---

## Round 0.2 · accepted · Accept

Dear authors, I congratulate you on the acceptance of this manuscript for publication.

·

Basic reporting

I have read the rebuttal letter and carefully reviewed the (marked-up) revised manuscript and Im satisfied with the authors response.
Nonetheless, the problem with the quality of the figures persists (they pixelate when zooming and thus are illegible), I guess that would be fixed at the production/proof reading stage.

I endorse the publication of this revised version

Experimental design

I endorse the publication of this revised version

Validity of the findings

I endorse the publication of this revised version

Additional comments

I endorse the publication of this revised version